# Transcriptomic Profiling and Pathway Analysis of Mesenchymal Stem Cells Following Low Dose-Rate Radiation Exposure

**DOI:** 10.3390/antiox12020241

**Published:** 2023-01-21

**Authors:** John E. Slaven, Matthew Wilkerson, Anthony R. Soltis, W. Bradley Rittase, Dmitry T. Bradfield, Michelle Bylicky, Lynnette Cary, Alena Tsioplaya, Roxane Bouten, Clifton Dalgard, Regina M. Day

**Affiliations:** 1Department of Pharmacology and Molecular Therapeutics, Uniformed Services University of the Health Sciences, 4301 Jones Bridge Rd., Bethesda, MD 20814, USA; 2Collaborative Health Initiative Research Program, Uniformed Services University of the Health Sciences, Bethesda, MD 20814, USA; 3Henry M. Jackson Foundation for the Advancement of Military Medicine, Bethesda, MD 20817, USA; 4Armed Forces Radiobiology Research Institute, Uniformed Services University of the Health Sciences, Bethesda, MD 20814, USA; 5The American Genome Center, Uniformed Services University of the Health Sciences, Bethesda, MD 20814, USA; 6Department of Anatomy, Physiology and Genetics, Uniformed Services University of the Health Sciences, Bethesda, MD 20814, USA

**Keywords:** radiation, low dose-rate, mesenchymal stem cells, human microvascular endothelial cells, RNAseq, gene regulation

## Abstract

Low dose-rate radiation exposure can occur in medical imaging, as background from environmental or industrial radiation, and is a hazard of space travel. In contrast with high dose-rate radiation exposure that can induce acute life-threatening syndromes, chronic low-dose radiation is associated with Chronic Radiation Syndrome (CRS), which can alter environmental sensitivity. Secondary effects of chronic low dose-rate radiation exposure include circulatory, digestive, cardiovascular, and neurological diseases, as well as cancer. Here, we investigated 1–2 Gy, 0.66 cGy/h, ^60^Co radiation effects on primary human mesenchymal stem cells (hMSC). There was no significant induction of apoptosis or DNA damage, and cells continued to proliferate. Gene ontology (GO) analysis of transcriptome changes revealed alterations in pathways related to cellular metabolism (cholesterol, fatty acid, and glucose metabolism), extracellular matrix modification and cell adhesion/migration, and regulation of vasoconstriction and inflammation. Interestingly, there was increased hypoxia signaling and increased activation of pathways regulated by iron deficiency, but Nrf2 and related genes were reduced. The data were validated in hMSC and human lung microvascular endothelial cells using targeted qPCR and Western blotting. Notably absent in the GO analysis were alteration pathways for DNA damage response, cell cycle inhibition, senescence, and pro-inflammatory response that we previously observed for high dose-rate radiation exposure. Our findings suggest that cellular gene transcription response to low dose-rate ionizing radiation is fundamentally different compared to high-dose-rate exposure. We hypothesize that cellular response to hypoxia and iron deficiency are driving processes, upstream of the other pathway regulation.

## 1. Introduction

Redox homeostasis has been defined as the balance between cellular generation of nonradical reactive oxygen species (ROS) and cellular antioxidant defenses [1,2,3,4]. Redox homeostasis has been demonstrated to contribute to normal cellular processes, including physiological redox signaling, oxidation and reduction of metabolic substrates, and oxidative phosphorylation [1,2,3]. However, oxidative damage to biological molecules can occur as the result of a redox imbalance, which can result in necrosis, apoptosis, and/or accelerated senescence [3,5]. Redox toxicity can occur as the result of exposure to oxidizing agents (ultraviolet and ionizing radiation, heavy metals, chemical toxins, etc.), from free radical production by normal physiological processes (mitochondrial oxidative phosphorylation, arachidonic acid metabolism, etc.), or from a deficit of antioxidant capacity of the cell [3,5,6,7]. Redox imbalances have been implicated in a wide variety of chronic diseases, such as cancer, cardiovascular disease, kidney disease, fibrotic remodeling diseases, and a variety of neurological diseases [2,6,8,9,10]. The cellular biological response to redox stress has been shown to depend upon the quantity, quality, duration, and total area of reactive species exposure [3,4,11,12,13,14].

Radiation damages biological macromolecules directly through energy deposition and indirectly through the generation of free radicals. Although free radicals have half-lives of milliseconds [14], they can induce chain reactions of oxidation or nitrosylation events, culminating in macromolecule modification and destruction [15]. Macromolecules oxidized following radiation exposure include DNA, proteins, lipids, and carbohydrates, each with specific downstream biological effects on cellular function [5,16,17,18,19]. DNA and proteins have been considered critical targets of radiation-induced oxidative damage [15,20,21], although all macromolecular damage has biological consequences. The biological response of a cell to radiation can be either repair and survival, several types of programmed cell death, necrosis, or accelerated senescence, depending on the qualities of radiation and specific antioxidant and repair defenses active in the cell [5,14,22,23].

The biological effects of radiation have been demonstrated to be dependent upon total dose, dose rate, and linear energy transfer values. High-dose-rate radiation is considered to be ≥0.1 Gy/min [24,25]. High-dose-rate ionizing radiation exposure can occur during cancer radiotherapy or due to accidental radiation exposure. A collection of potentially life-threatening syndromes—Acute Radiation Syndrome (ARS) and Delayed Effects of Acute Radiation Exposure (DEARE)—can occur from high-dose radiation exposure, depending upon the total dose of radiation, the dose rate, and the area of radiation exposure [26,27]. In contrast, low dose-rate irradiation has been defined as between 0.006–0.1 Gy/h [28]. Low dose-rate radiation exposure can occur in medical imaging, as a result of low levels of ionizing radiation in the environment, or as a hazard of space travel [29]. In contrast with high-dose-rate radiation exposure, the effects of chronic low-dose radiation are associated with Chronic Radiation Syndrome (CRS). The characteristics of CRS are not immediately life-threatening, but result in alterations in environmental sensitivity. Early events in CRS include increased olfactory and taste thresholds, sensitivity to vibration, and changes in systemic immunity [30]. Secondary effects of chronic radiation exposure have been observed in the circulatory and digestive systems, although these have been shown to spontaneously revert when radiation exposure ends [30]. Cardiovascular and neurological diseases are of concern for long-distance space travel [31,32]. Cancer is believed to be a serious risk of low-dose radiation, but the response of individuals to low-dose radiation for the prediction of cancer risk is still poorly defined [29].

In general, acute high-dose radiation exposure induces greater macromolecular damage and results in greater suppression of cell proliferation compared with low dose-rate radiation [33]. Cellular responses to high dose-rate irradiation include the immediate activation of DNA damage response and DNA repair pathways, upregulation of cell cycle inhibitors, ER stress response, and the unfolded protein response pathway [21,23]. These responses have been demonstrated to follow the robust activation of specific enzymes and transcription factors responsive to oxidative stress and DNA damage [14,34]. In comparison, the effects of low dose-rate radiation are less well understood [29,30,35].

Radiation effects have been shown to have cell-type and tissue-type specificity [36]. Mesenchymal stem cells (MSCs) are multipotent adult stem cells, originating in the bone marrow, with the capacity to enter through the circulation, migrate to injured tissue, engraft, and differentiate into different phenotypes depending upon their environment [37,38]. MSCs were previously demonstrated to be relatively resistant to radiation-induced damage, requiring 2 Gy to reduce the surviving fraction of cells to 37% [39]. 10 Gy was demonstrated in MSCs to reduce the surviving population to <1% [39]. Because of the importance of MSC in repair following tissue injury, we investigated the effects of chronic low dose-rate (0.66 cGy/h) radiation on MSCs. We confirmed our findings in primary human lung microvascular endothelial cells. Our findings indicate that the pathways regulated by chronic low dose-rate radiation differ greatly compared with those regulated by acute high dose-rate radiation exposure.

## 2. Methods

### 2.1. Reagents

Chemicals and reagents were purchased from MilliporeSigma (St. Louis, MO, USA) except where indicated.

### 2.2. Cell Culture and Irradiation

Human bone marrow mesenchymal stem cells (MSCs) were purchased from Lonza (Morristown, NJ, USA) and cultured in mesenchymal stem cell growth medium (Lonza). Human lung microvascular endothelial cells (HLMVECs) were purchased from Cell Applications (San Diego, CA, USA) and cultured on plates treated with endothelial cell attachment factor in Microvascular Endothelial Cell Growth Medium (Cell Applications). Cells were grown and irradiated in a humidified environment of 5% CO_2_/95% air at 37 °C, according to the manufacturer’s instructions. Cells were used within seven passages for all experiments. For irradiations, cells were plated in 25 ml flasks or in Lab-Tek Flaskette Chamber slides (ThermoFisher Scientific, Waltham, MA, USA), and grown to 60–70% confluence. Cells were irradiated in the Armed Forces Radiobiology Research Institute (AFRRI) Low Level Cobalt Facility at 0.66 cGy/h to reach a total irradiation of 0.3, 0.7, 1, or 2 Gy. Dosimetry for Low Level Cobalt Facility was performed by preliminary mapping of irradiation field inside the cell culture incubator. The irradiation field from the bare source was measured using an A12 ion chamber calibrated at ADCL, University of Wisconsin. The dose rate was measured at the position of each flask (21 points of measurements, 7 flasks on 3 shelves) under the same conditions as actual cell irradiation conditions. To achieve desired dose rate, the 4-times attenuator was mounted and field uniformity was measured at the same positions with ion chamber PTW TN 75 cc. The dose rate on a date of mapping of the field was measured as 0.6646 cGy/h with field uniformity 0.28%. Control dishes were cultured in parallel in a separate incubator without radiation exposure. The 10 Gy irradiation was performed on cells at 70–90% confluence using an RS2000 Biological Irradiator (Rad Source Technologies, Alpharetta, GA, USA) at a dose rate of 1.15 Gy/min (160 kV, 25 mA) for a total dose of 10 Gy, as previously described with previously described dosimetry [40].

### 2.3. Cellular and Nuclear Morphology

Cells were plated on Nunc Lab-Tek flask on a slide (ThermoFisher Scientific). Cell irradiations were initiated once cells reached 60–70% confluent, and analyses were conducted in triplicate after 7 and 14 days of chronic low dose-rate radiation exposure, or at 3 and 7 days after acute high-dose-rate radiation exposure. Dishes were washed twice with phosphate-buffered saline (PBS), and fixed with 3.7% formaldehyde in PBS for 5 min at room temperature, washed twice more with PBS, and then mounted on slides in Prolong Gold antifade reagent with DAPI (Invitrogen, Eugene, OR, USA). Cells were imaged as for DAPI staining. All nuclei were analyzed in 5 random images, at least 100 nuclei per slide and three slides per time point, by a researcher blinded to treatment groups [40].

### 2.4. Gamma-H2AX Immunohistochemistry

Cells were fixed with 4% paraformaldehyde for 20 min at room temperature, then washed in PBS for 5 min, 3 times. Next, cells were permeabilized using 70% ethanol for 5 min at room temperature. Cells were blocked for 1 h in 5% normal donkey serum in PBS and then incubated in anti-γ-H2AX (#9718S, Cell Signaling, Danvers, MA, USA), 1:400 dilution in blocking solution for 1 h. Blocking and primary incubation were performed in a humidified chamber at 37 °C. Cells were then washed 3× in PBS before incubating for 1 h at room temperature in secondary antibody (Thermo Fisher, A21206), 1:2000 in PBS. Cells were again washed 3× in PBS and coverslipped with Pro-Long Gold antifade with DAPI. A Nikon Eclipse Ti microscope with a Nikon DSRi2 camera was used for digital images (Nikon Instruments, Inc., Melville, NY, USA). All images were obtained using the same exposure conditions to avoid false positives and false negatives in the scoring and images. Scoring was performed by a researcher blinded to the conditions. γ-H2AX foci were counted in DAPI-positive cells by scoring all cells within a field, and counting at least 100 cells per slide.

### 2.5. Quantitative PCR Analysis

Immediately following irradiation, total RNA was isolated from MSC or HLMVEC using the RNeasy Mini Kit with on-column DNase digestion (Qiagen, Valencia, CA, USA) according to manufacturer’s protocol. RNA was quantified spectroscopically (ND-1000 Spectrophotometer, Nano-Drop, Wilmington, DE, USA) and 1.0 µg was reverse transcribed using iScript cDNA synthesis kit (Bio-Rad, Hercules, CA, USA), according to the manufacturer’s protocol. RT-qPCRs were performed in technical duplicates using iTaq™ Universal SYBR Green Supermix (Bio-Rad), on a CFX96 Touch Real-Time PCR Detection System (Bio-Rad) as described [40]. Primers for qRT-PCR were designed using NCBI/Primer-BLAST and purchased from Integrated DNA Technologies (Coralville, IA, USA). Forward and reverse primer sequences are shown in Table 1. Relative gene expression to the reference genes was calculated using the ΔΔCq method using CFX Maestro software, 2.0 (Bio-Rad) [41,42].

### 2.6. Western Blotting

Immediately following irradiation, cells were lysed for 20 min at 4 °C in RIPA buffer (ThermoFisher Scientific) with protease and phosphatase inhibitors (#A32953 and #A32957, ThermoFisher Scientific). Lysates were centrifuged at 7000 RCF for 7 min at 4 °C. Proteins were separated by polyacrylamide gel electrophoresis and transferred to nitrocellulose membranes (MilliporeSigma) as previously described [40]. Nitrocellulose membranes were blocked in Tris-buffered saline with 5% BSA for 1.5 h. Primary antibodies were diluted in 5% BSA in Tris-buffered saline: β-actin (MilliporeSigma #A1982, 1:5000); DUSP1 (Cell Signaling #35217; 1:1000); total and phosphorylated p44/42 MAPK (Cell Signaling, Danvers, MA, USA, #4696 and #4370; 1:2000); total and Ser473 phosphorylated Akt (Cell Signaling #2920 and #4060; 1:2000); EGR1 (Cell Signaling #4154; 1:1000); full-length and cleaved caspase-3 (Cell Signaling #14220 and #9661; both 1:1000). Conjugated secondary antibodies (LI-COR, Lincoln, NE, USA; 1:10,000) were used for detection using the Odyssey system (LI-COR). β-actin was used as a loading control and for normalization of sample concentrations.

### 2.7. Transcriptome Profiling by RNA Sequencing

Immediately following irradiation, total RNA was isolated from MSC using the RNeasy Mini Kit with on-column DNase digestion (Qiagen) according to manufacturer’s protocol. RNA was quantified spectroscopically (ND-1000 Spectrophotometer, Nano-Drop, Wilmington, DE, USA). The total RNA integrity was assessed using automated capillary electrophoresis with a Fragment Analyzer (Roche, Pleasanton, CA, USA). For all samples with an RNA quality indicator (RQI) > 8.0, a total of >75 ng RNA was used as the input for library preparation using the TruSeq Stranded mRNA Library Preparation Kit (Illumina, San Diego, CA, USA). The sequencing libraries were quantified by Real-Time PCR on a Roche LightCycler 480 Instrument II using a KAPA Library Quantification Kit for NGS (Kapa, Wilmington, MA, USA). The size distribution was assessed by automated capillary-based gel electrophoresis with a Fragment Analyzer to confirm the absence of free adapters or adapter dimers. The sequencing libraries were pooled and sequenced on a NovaSeq 6000 Sequencer (Illumina) using a NovaSeq 6000 SP Reagent Kit (300 cycles) within one flowcell lane using an XP workflow with 101 + 8 + 8 + 101 cycle parameters with paired-end reads of 75 bp in length. Raw sequencing reads were demuxed using bcl2fastq2 (v2.20) and aligned to the human reference genome (hg38) with MapSplice (v2.2.2) [43]. Gene-level quantification was performed with HTSeq (v0.9.1) [44] against GENCODE (v28) basic gene annotations. Read alignment statistics and sample quality features were calculated with Samtools and RseQC [45,46,47]. Sequencing quality was verified by manual inspection of sample-wise characteristics: total reads, mapping percentages, pairing percentages, transcript integrity number (TIN), 5′ to 3′ gene body read coverage slopes, and ribosomal RNA content [48]. The transcript abundance quantitation data were deposited in the NCBI Gene Expression Omnibus (GSE222541). Differential expression analysis was performed with DESeq2 (v1.16.1) [49] on raw gene counts. We defined significant differentially expressed genes (DEGs) between irradiated and control samples as those with a False Discovery Rate (FDR) q-value < 0.05, an absolute fold change > 1.5 (i.e., |log_2_ (fold-change)| > 0.585), and mean transcripts per million (TPM) ≥ 1 across samples.

### 2.8. Gene Ontology, Pathway Enrichment Analysis, and Heat Map Construction

Gene Ontology (GO) and Kyoto Encyclopedia of Genes and Genomes (KEGG) pathway analyses were performed using the Database for Annotation, Visualization, Integrated Discovery (DAVID), version 6.8, with the medium classification stringency, an enrichment threshold of 0.05, and the Bonferroni method of adjustment for multiple testing (Laboratory of Human Retrovirology and Immunoinformatics, Frederick, MD, USA) [50,51]. GO was also performed using Gene Ontology enRIchment anaLysis and visuaLizAtion tool (GOrilla), version 4.1 (http://cbl-gorilla.cs.technion. ac.il/accessed on 15 October 2022) [52,53]. Venn diagrams were constructed using Venny, version 2.1 (Juan Carlos Oliveros, BioInfoGP Service, Centro Nacional de Biotecnologia, Madrid, Spain; https://bioinfogp.cnb.csic.es/tools/venny/ accessed on 9 July 2022). Pathway interconnection were determined using Metascape (https://metascape.org accessed on 15 November 2022 [54]).

### 2.9. Statistics

Statistical analyses of assays were performed using Graphpad Prism 7 (San Diego, CA, USA) or Excel. For RNA-seq and qPCR analysis, one-way ANOVA with a post-test analysis was used for comparing multiple data sets. For Western blot analysis, two-way ANOVA with either Tukey’s or Sidak’s post hoc tests for multiple comparisons were used.

## 3. Results

### 3.1. Low Dose-Rate Radiation Effects on Cellular Morphology, Apoptosis, and Double-Stranded DNA Breaks in Mesenchymal Stem Cells (MSC)

Our previous studies showed that high-dose/high-dose-rate (10 Gy/0.989–1.15 Gy/min) X-ray irradiation primarily induces accelerated senescence in primary pulmonary artery endothelial cells (PAEC), primary human lung microvascular endothelial cells (HLMVEC), and primary mesenchymal stem cells (MSCs) [22,23,40]. We compared irradiation of MSC at low- and high-dose-rates. The morphology of MSCs following 0.66 Gy/h ^60^Co radiation showed that the cells maintained a consistent morphology after 7 days (~1 Gy) and 14 days (~2 Gy) (Figure 1A). Additionally, cells appeared to increase in density over the time course of the experiment. In contrast, MSCs exposed to 10 Gy X-ray irradiation (1.15 Gy/min) displayed flattened “fried egg” morphology with increased cellular area and reduced cell numbers, consistent with cellular senescence, at 2 weeks post-irradiation (Figure 1A).

We previously found that low levels of apoptosis were present in 10 Gy (0.989–1.15 Gy/min) X-ray irradiated PAECs, HLMVECs, and MSCs, although in most cases the increase in apoptosis did not reach significance compared with control levels [22,23,40]. We investigated the induction of apoptosis in low- and high-dose-rate irradiation in MSCs using nuclear morphological analysis (Figure 1B) [40]. Nuclear blebbing is consistent with late apoptotic events. We did not observe any significant increase in nuclear morphological changes in any of the irradiated cells at the time points examined.

We examined DNA damage in the cells after 1 and 2 Gy (0.66 cGy/h, ^60^Co) or 10 Gy (1.15 Gy/min, X-ray) exposures. DNA damage initiates signaling pathways that result in the phosphorylation of serine 139 on histone H2AX to form γ-H2AX, which is present in complexes surrounding double-stranded DNA breaks [55]. Immunohistochemistry for γ-H2AX complexes in the nuclei of MSCs following low- and high-dose-rate radiation exposure showed a significant increase in foci at 3 days and 2 weeks following 10 Gy/1.15 Gy/min X-ray irradiation (Figure 1C). Interestingly, exposure to low dose-rate irradiation did not show a significant increase in γ-H2AX foci compared with basal levels.

### 3.2. Genome-Wide Transcriptional Responses to Low Dose-Rate Radiation

To expand the understanding of overall gene expression changes in primary MSCs in response to low dose-rate radiation, we used comprehensive transcriptome profiling by RNA-seq. Gene expression profiles from sham-irradiated (control) MSCs were compared with MSCs irradiated at 0.66 cGy/h for 1 Gy total (~1 week) and 2 Gy total (~2 weeks). Comparative differential expression analysis identified 862 genes differentially expressed between 1 Gy irradiation samples and matched controls (q-value < 0.05, absolute fold change > 1.5) (Appendix A). For 2 Gy irradiation, comparative differential expression analysis identified 725 differential genes (q-value < 0.05, absolute fold change > 1.5) (Appendix A). A heatmap of the differentially expressed genes (DEGs) of all samples is shown in Figure 2A. At 1 Gy, 472 genes were downregulated compared with the matched control, and 390 genes were upregulated. At 2 Gy, 427 genes were downregulated compared with the matched control, and 298 genes were upregulated. A comparison of the gene sets from 1- and 2-Gy-regulated genes showed an overlap of ~39% of genes that were regulated at both doses of radiation (Figure 2B).

GO analyses were focused on terms relevant to cellular biological processes (BPs) and not disease states. The BP graphs show clusters of pathways with enrichment scores ≥ 1.6 (Figure 3, Appendix A). DAVID analysis showed that following 1 Gy low dose-rate irradiation (1 week, 0.66 cGy/h), the largest changes (both up- and down-regulation) in BP terms were vasoconstriction and blood pressure, metabolic processes (including glucose, cholesterol, fatty acid/lipid metabolism, and cellular response to starvation), proliferation, cellular response to hypoxia/reactive oxygen species and iron ion responses, apoptosis, and adhesion, and extracellular matrix modification (Figure 3A,B). In contrast, following 2 Gy low dose-rate irradiation (2 weeks, 0.66 cGy/h), DAVID analysis showed positive regulation of proliferation, continued metabolic processes (cholesterol and fatty acid biosynthesis/metabolism, response to starvation), continued cellular response to hypoxia, extracellular matrix modification (especially collagen), and apoptotic signaling (Figure 3C,D). Metascape (https://metascape.org accessed on 10 August 2022) [54] was used to create an image of clustered GO terms present in 1300 genes with the lowest q-value (up- and down-regulated, from both 1 Gy and 2 Gy) (Figure 3E). Relationships were identified between the pathway functions, notably the signaling genes for MAPK, protein phosphorylation and enzyme-linked receptors with genes that regulate cellular adhesion and locomotion, vascular development, tissue morphogenesis, and skeletal system development. Additional links were identified between regulation of genes for extracellular matrix organization, supramolecular fiber organization, and overall changes in genes encoding the core proteins making up the extracellular matrix (ECM; NABA core matrisome) and the ECM-associated proteins (NABA-matrisome-associated). GOrilla analysis of the ranked DEGs with q value ≤ 10^−5^ also identified enrichment in cellular processes, including cholesterol, lipid, sterol, and alcohol metabolism, protein catabolism, polysaccharide metabolism, responses to hormones and oxidative stress, extracellular matrix reorganization and cell motility, and developmental processes, including cell differentiation, and regulation of tissue remodeling, including vascular smooth muscle and bone remodeling (data not shown). GOrilla analysis also showed major changes in pathways regulating proliferation, inflammation, and programmed cell death (data not shown).

### 3.3. Focused Heatmap Analysis of Chronic Low-Dose Radiation Gene Regulation

According to the GO analysis by DAVID and GOrilla, we evaluated gene regulation in the pathways and processes found to be most affected by chronic low-dose radiation. The focused analyses included genes selected by the DAVID and GOrilla, and included additional genes that we curated through literature searches for genes involved in each process that were also regulated in our study. qPCR and/or Western blotting was used to validate the pathways identified by RNAseq.

#### 3.3.1. Alteration of Cellular Metabolism

Chronic low-dose radiation affected a number of metabolic pathways in the MSCs, including downregulation of cholesterol synthesis, upregulation of glycolysis over oxidative phosphorylation, and a reduction in fatty acid biosynthesis and modification (Figure 4A). qPCR was performed to validate at least one gene in each pathway (Figure 4B). Both 1 Gy and 2 Gy low dose-rate radiation showed downregulation of almost all genes encoding cholesterol synthesis enzymes, including for the synthesis of squalene from acetyl-CoA and for the synthesis of cholesterol from squalene (Figure 4A, left panel). These genes include the enzyme for the initiation of cholesterol synthesis (acetyl-CoA acetyltransferase 2, ACAT2), the rate-limiting enzyme (3-hydroxy-3-methylglutaryl-CoA synthase 1, HMGCS1), through to the final enzyme in the pathway (24-dehydrocholesterol reductase, DHCR24). In addition to the downregulation of cholesterol synthesis enzymes, we also observed the suppression of two major regulators of cholesterol biosynthesis: insulin-induced gene (INSIG1) and sterol regulatory element binding transcription factor-1 and -2 (SREBF1 and SREBF2). Finally, we observed downregulation of Niemann-Pick Type C disease 1 (NPC1), which regulates intracellular cholesterol transport and esterification.

Low-dose chronic radiation also resulted in the regulation of metabolic pathways that favored glucose metabolism over mitochondrial oxidative phosphorylation (Figure 4A, middle panel). Genes associated with increased glycolysis were upregulated: 3-phosphate dehydrogenase (GAPDH), pyruvate dehydrogenase kinases (PDK1, 3, and 4), aldo-keto reductase family 1 member C3 (AKR1C3), leptin (LEP), phosphofructo-2-kinase/fructose-2,6-bisphosphatase 3 (PFKFB3), proprotein convertase subtilisin (PCSK9), FOXO1, and NUAK2. PCSK9, which can negatively regulate glucose metabolism, was decreased five-fold. PDK1, -3, and -4, were each upregulated ~two-fold, inhibiting pyruvate dehydrogenase, and reducing the production of acetyl-coenzyme A from pyruvate. The upregulation of the PKDs can result in decreased activity of the tricarboxylic acid (TCA) cycle, decreased oxidative phosphorylation, and increasing the production of lactate as a final downstream function of glycolysis. We also observed an increase in lactate dehydrogenase, which catalyzes the conversion of pyruvate to lactate, again suggesting that pyruvate is being diverted away from the TCA cycle. FOXO1, a transcription factor responsible for increased gluconeogenesis, was reduced. Nu [novel] AMPK-related protein kinase-2 (NUAK2), which is responsive to increased AMP/decreased ATP, low glucose, and oxidative or endoplasmic reticulum stress, signals to suppress cell death by glucose starvation, was increased.

There was a general decrease in genes encoding enzymes for fatty acid (FA) metabolism and processing. Low-density lipoprotein receptor (LDLR), which can take up lipids from the environment, was decreased. There were decreases in folliculin (FLCN) and folliculin-interacting proteins (FNIP1 and FNIP2) regulators of AMP-dependent protein kinase (AMPK), a master regulator of FA metabolism, antioxidant responses, and mitochondrial and lysosome biogenesis. Decreases were observed in a number of types of FA acid synthesis enzymes: fatty acid synthase (FASN), a central regulator of lipid metabolism; acetyl-CoA carboxylase, which catalyzes the rate-limiting step in long-chain FA biosynthesis; acyl-CoA synthetase family member 2 (ACSF2), which enables medium-chain FA ligase activity; desaturase enzymes (FADS1 and 2), which produce highly unsaturated FA (HUFA); and stearoyl-coenzyme A desaturase (SCD), which synthesizes monounsaturated FA. Enzymes for the processing of FA were decreased, including for FA desaturation (FA desaturase-1 and 2, FADS1 and 2) and transport (FA binding protein 3, FABP3). Additionally, enzymes for FA degradation were also reduced: HADH, ASAH1, PLA1A, and GBA.

#### 3.3.2. Regulation of Proliferation and Cell Division

We observed mixed regulation of genes related to cell proliferation. Focused heatmaps of proliferation and cell cycle genes are shown in Figure 5A; three of the regulated genes were validated by qPCR (Figure 5B). However, the cell numbers and morphology determined by light microscopy suggested that the MSCs continued to proliferate over the course of the 1 and 2 Gy exposures, without significant apoptosis or accelerated senescence (see Figure 1). We observed upregulation of proliferation-inducing genes. Upregulated growth factors included fibroblast growth factor 1 (FGF1), transform in growth factor A (TGFA), and vascular endothelial growth factor A (VEGFA). Upregulated transcription factors included Odd-skipped related transcription factor (OSR1) and c-Jun (JUN). We also observed downregulation of other growth factors, such as colony stimulating factor (CSF1) and pleiotrophin (PTN), as well as some transcription factors, such as transcription factor AP4 (TFAP4). There was also mixed regulation of factors that regulate apoptosis, including the upregulation of Baculovirus inhibitor of apoptosis repeat containing 5 (BIRC5), which suppresses apoptosis and promotes proliferation, but also the downregulated proteins that inhibit apoptosis, such as SFRP4 and IFIT3.

There was a general downregulation of cell-cycle regulatory proteins, including cyclin-dependent kinase 1 (CDK1), cyclin-dependent kinase 4 (CDK4), cyclin A2 (CCNA2), cyclin-dependent kinase-like 1 (CDKL1), cyclins B1 and 2 (CCNB1 and 2), and cell-division-cycle-associated A8, 20 and 25B (CDCA8,20, 25B). Proteins interacting with the chromosome, centromere, and mitotic spindle were also downregulated, including kinesin family members C1 and C2 (KIFC1, 2), nucleolar-spindle-associated protein (NUSAP1), centrosomal protein 55 (CEP55), centromere protein F (CENPF), and condensin subunit CAP g (NCAPG). These proteins are known to have roles in centrosome stabilization, chromosome condensation, and mitotic spindle formation, all required for cell division.

Interestingly, the downregulation of these proteins occurred without the induction of apoptosis or accelerated senescence (See Figure 1). The phosphorylation of Akt and p42/p44 MAPK are associated with proliferation and cell survival, and we observed significant increases in both phosphorylated Akt and p42/p44 MAPK in the 1 and 2 Gy exposures (Figure 5C). This suggests that the mixed regulation of proliferation and cell cycle genes favors cell survival and proliferation, or potentially, cell cycle arrest without senescence.

#### 3.3.3. Regulation of Apoptosis, Cell Death, and Autophagy

A variety of forms of cell death and autophagy are induced in cells in response to high-dose acute ionizing radiation, and we previously observed the upregulation of pro-apoptotic and pro-senescence pathways [5,40]. In contrast, we did not observe significant levels of apoptosis at any time points following chronic low-dose radiation (see Figure 1), and GO analysis showed apoptotic pathway regulation predominantly favoring the inhibition of apoptosis by a number of mechanisms (Figure 6). The regulation of this group of genes was confirmed by qPCR of EGR1 and DUSP1 and Western blotting of Egr1 and DUSP1 (Figure 6B,C).

Gene regulation was observed in the pathways for intrinsic and extrinsic apoptosis, p53-pathway-regulated apoptosis, and Wnt signaling-induced apoptosis. Two pro-apoptotic master regulators were downregulated: forkhead box protein O1 (FOXO1) and nephroblastoma-overexpressed protein (NOV). Three anti-apoptotic regulators were upregulated: early growth response 1 (EGR1), leptin (LEP), and eukaryotic elongation factor 1A2 (EEF1A2). There was most notably a reduction of a number of genes in the p53-mediated pathway of apoptosis, including DNA-damage-regulated autophagy modulator 1 (DRAM1), interferon gamma-inducible protein 16 (IFI16), breast cancer suppressor protein (BRCA1), NADH:ubiquinone oxidoreductase subunit A13 (NDUFA13), maternal embryonic leucine zipper kinase (MELK), p53-induced death domain protein 1 (PIDD1), and oxidative-stress-induced growth inhibitor 1 (OSGIN1). Several pro-apoptotic genes in the extrinsic/death-receptor-initiated apoptosis pathway were downregulated: Toll-like receptor 3 (TLR3), death-associated protein kinase 2 (DAPK2), Huntingtin-interacting protein 1 (HIP1), and sequestosome-1 (SQSTM1). At the same time, two anti-apoptotics for the extrinsic apoptotic pathway were upregulated: tumor necrosis factor receptor superfamily member 11B (TNFSF11B) and family member 10D (TNFRSF10D).

Interestingly, we did observe regulation of pro-apoptotic pathways associated with DNA-damage-, oxidative-stress-, or hypoxia-induced apoptosis. Examples of these included downregulation of anti-apoptotic genes for secreted frizzled related proteins 2 and 4 (SFRP2 and 4), 24-decydrocholesterol reductase (DHCR24), DNA-damage-induced apoptosis (DDIAS), and chitinase-3-like protein 1 (CH13L1), as well as increased expression of pro-apoptotic dual-specificity protein phosphatase 1 and 6 (DUSP1 and 6), immediate early response 3 (IER3), family with sequence similarity 162 member A (FAM162A), osmotic-stress-resistance protein (OSR), and BCL2/adenovirus E1B 19 kDa protein-interacting protein 3 (BNIP3). As stated previously, we did not observe significant apoptosis in the cells, suggesting that the overall effect of these mixed regulations favored cell survival.

#### 3.3.4. Regulation of Pathways for Extracellular Matrix, and Cell Attachment and Migration

GO analysis of the gene changes showed significant radiation-induced alterations in pathways related to synthesis, breakdown, and organization of the extracellular matrix (ECM), as well as for cell attachment to the extracellular matrix (Figure 7A). Two genes from these pathways, keratin 34 (KRT34) and neuregulin 1 (NRG1), were validated by qPCR (Figure 7B).

Changes in gene expression related to proteins of the ECM were characterized by upregulation of genes associated with wound repair and/or fibrosis. The upregulated genes included markers of fibrotic remodeling (collagen 4A1 and 5A1 [COL4A1, COL5A1], prolyl 4-hydroxylase subunit alpha 1 [P4H1], procollagen-lysine, 2-oxoglutarate 5-dioxygenase 2 [PLOD2], lysyl oxidase-like 4 [LOXL4], prolyl 3-hydroxylase 2 [P3H2], a disintegrin and metalloproteinase with thrombospondin motifs 2 [ADAMTS2]) as well as known drivers of fibrosis (hyaluronic acid synthase 1 and 2 [HAS1, HAS2], prolyl 4-hydroxylase subunit alpha 2 [P4H2], lysyl oxidase [LOX], ADAMTS6, and procollagen C-endopeptidase enhancer 1 [PCOLCEL]). The downregulated genes also included some other markers of fibrosis (vitronectin [VTN], fibrillin 1 [FBN1], cartilage oligomeric matrix protein [COMP], integrin subunit alpha 4 [ITGA4], secreted frizzled related protein 2 [SFRP2], proline and arginine rich end leucine rich repeat protein [PRELP], stearoyl-CoA desaturase 1 [SCD1], and thrombospondin 3 [THBS3]) as well as genes with antifibrotic function (disintegrin and metalloproteinase domain-containing protein 12 [ADAMS12], and cathepsins D and K [CTSD, CTSK, two proteases]). These changes were accompanied by decreased expression of some proteins for ECM breakdown and decreased proteins that functioned to maintain or increase barrier function: filaggrin (FLG), interleukin 32 (IL32), intercellular adhesion 1 (ICAM1), wingless/integration 1 (WNT1) inducible-signaling pathway protein 2 (WISP2), and trophinin-associated protein (TROAP). The combination of these alterations of proteins of the ECM could function to increase the matrix stiffness.

The regulation of pathways for cell adhesion included a number of genes predicted to increase cellular motility and/or invasiveness: neuregulin 1 (NRG1), fibroblast growth factor 1 (FGF1), WNT family member 5B (WNT5B), heparin-binding epidermal growth factor-like growth factor (HBEGF), nestin (NES), semaphoring 7A (SEMA7A), endothelin 1 (EDN1), cysteine-rich angiogenic inducer 61 (CYR61), erythrocyte membrane protein band 4.1-like B4 (EPB41L4B), cluster of differentiation 274 (CD274, also known as programmed cell death 1 ligand 1), desmin (DES), sushi repeat-containing protein X-linked 1 (SRPX2), transient receptor potential cation channel subfamily M member 8 channel-associated factor 2 (TCAF2), and glutamine gamma-glutamyltransferase 2 (TGM2). Additional changes in cellular intermediate fillaments included increased expression of a number of keratin (KRT) proteins, including KRT7, 8, 14, 16, 18, 34, 80, and 81, which also can increase cellular stiffness and promote motility. We did not observe upregulation of any proteins associated with inhibition of motility, but we did observe the downregulation of several proteins that would increase cellular adhesion: secreted phosphoprotein 1 (SPP1), desmoplakin (DSP), and metastasis suppressor protein 1 (MTSS1).

#### 3.3.5. Regulation of Pathways for Vascular Constriction and Inflammation

GO analysis of the transcriptome revealed alterations in pathways for vascular constriction and inflammation (Figure 8A). Three of these genes were validated using qPCR: OXTR, HMOX1, and NPR3, which showed similar alterations as the transcriptomic data (Figure 8B). The regulation of vascular constriction pathways showed an increase in the number of genes associated with increased vasoconstriction: 5-hydroxytrypamine receptor 2A (HTR2A), adrenoreceptor alpha 1B and 1D (ADRA1B and 1D), oxytocin receptor (OXTR), prostaglandin–endoperoxide synthase 2 (PTGS2, also called cyclooxygenase-2), gap junction alpha 5 (GJA5), leptin (LEP), apelin (APLN), and endothelin 1 (EDN1), which has a specific role in pulmonary hypertension. At the same time, there was a decrease in expression of several genes associated with vasodilation: superoxide dismutase 2 (SOD2) and heme oxygenase 1 (HMOX1).

Radiation-induced changes in expression of inflammatory factors and activators showed mixed up- and down-regulation of pro- and anti-inflammatory factors. Upregulated anti-inflammatory factors included: secreted and transmembrane 1 (SECTM1), osteoprotegerin (TNFRSF11B), tumor necrosis factor receptor superfamily member 10D (TNFRSF10D), leukemia inhibitory factor (LIF), semaphoring 7A (SEMA7A). At the same time, we observed downregulation of some pro-inflammatory factors: TNFRSF21, apolipoprotein E (APOE), nephroblastoma-overexpressed protein (NOV), and PDZ-binding protein (PBK).

#### 3.3.6. Regulation of PATHWAYs for Cell Response to Hypoxia and Iron Homeostasis

GO analysis revealed the regulation of pathways related to cellular response to hypoxia as well as to iron homeostasis and iron-binding proteins (Figure 9A). Four of the genes in these groups were validated by qPCR (Figure 9B). Interestingly, the regulation of genes to hypoxia included a number of genes that are also regulated by redox stress [56], and a number of genes related to iron homeostasis and iron-binding can also be regulated by redox stress [57].

Surprisingly, we observed downregulation of a number of enzymes that would mitigate redox stress, including superoxide dismutase 1 and 3 (SOD1, 3) and glutathione peroxidase 1 and 4 (GPX1, 4). We also observed the downregulation of the oxidative stress-activated transcription factor nuclear factor erythoid-derived 2-like 1 and 2 (NFE2L1 and 2), as well as its regulator, kelch-like ECH-associate protein 1 (KEAP1). In contrast, we observed an upregulation of almost 20 genes previously shown to be regulated by hypoxia-inducible factor-1α (HIF-1α): pyruvate dehydrogenase kinase 4 (PDK4), (FAM162A), (MGARP), stanniocalcin-1 (STC1), procollagen-lysine, 2-oxoglutarate 5-dioxygenase 2 (PLOD2), lactate dehydrogenase-A (LDHA), endothelin 1 (EDN1), BCL2/adenovirus E1B 19 kDa protein-interacting protein 3 (BNIP3), angiopoietin-like 4 (ANGPTL4), vascular endothelial growth factor A (VEGFA), solute carrier family 2, facilitated glucose transporter member 8 (SLC2A8), aldehyde oxidase (AOX1), aquaporin-1 (AQP1), apolipoprotein L domain-containing protein 1 (APOLD1), potassium two-pore-domain channel subfamily K member 3 (KCNK3), sodium calcium exchanger 1(NCX1 or SLC8A1), prostaglandin–endoperoxide synthase 2 (cyclooxygenase, PTGS2), and neuron-derived neurotrophic factor (NDNF). These genes have a variety of functions including the regulation of glycolysis, cholesterol and lipid biosynthesis, increased proliferation, inhibition of apoptosis, extracellular matrix modification, and cell motility. The regulation of hypoxia-responsive membrane channels, together with a large group of genes regulated by HIF-1α, strongly suggest the induction of cellular adaptation to a hypoxic environment.

Iron-regulated pathways were also identified by GO analysis. Two major proteins for iron storage were downregulated: ferritin light chain (FTL) and ferritin heavy chain 1 (FTH1). At the same time, there was a downregulation of a number of iron-dependent enzymes, including a number of enzymes involved in fatty acid and cholesterol metabolism: fatty acid desaturase 1 and 2 (FADS1, FADS2), lanosterol 14-alpha demethylase (CYP51A1), hydroxysteroid 17-beta dehydrogenase-14 and 11-beta dehydrogenase 1 (HSD17B14 and HSD11B1), 7-dehydrocholesterol reductase (DHCR7), cytochrome B5 reductase-like (CYB5RL), stearoyl –CoA-desaturase 1 and 5 (SCD, SC5D), and flavin containing dimethylaniline monooxygenase 3 and 4 (FMO3, FMO4). Besides these, four genes were regulated with iron valence modification activity: cytochrome B reductase 1 (CYBRD1), six-transmembrane epithelial antigen of prostate 1(STEAP1), cytoglobin (CYGB), and ferric-chelate reductase 1 (FRRS1). Interestingly, three of the four upregulated genes were all related to cellular response iron deficiency: family with sequence similarity 162 member A (FAM162A), endothelin 1 (EDN1), and stanniocalcin 1 (STC1). Together, these gene regulation patterns suggests cellular responses to iron deficiency and downstream signaling.

### 3.4. Effect of Chronic Low Dose-Rate Irradiation on Primary Human Lung Microvascular Endothelial Cells

The vascular endothelium is hypothesized to be a primary mediator of radiation injuries [58]. We therefore investigated the effect of low dose-rate radiation on primary human lung microvascular endothelial cells in culture using the same targets identified in MSCs (Figure 10). We found similar regulation for genes in cholesterol and fatty acid biosynthesis (HMGCS1 and MSMO1), for cell survival and suppression of accelerated senescence (EGR1 and CDKN1A and MAPK and Akt activation), for reduced antioxidant signaling (SOD2), and we observed a trend for extracellular matrix modification (COL1A2).

## 4. Discussion

The potential for accidental radiation exposure has increased with the increasing use of medical and industrial radiation, and with the potential increase in nuclear energy generation and the potential for military use of radioactive weapons. Current proposals for space travel will also result in chronic radiation exposure. Therefore, an increased understanding of the cellular effects of chronic low dose-rate radiation is needed. Here, we demonstrate in primary human MSCs that exposure to 0.3–2 Gy gamma radiation (0.66 cGy/h) did not result in growth arrest, the induction of significant apoptosis, or accelerated senescence. GO analysis of the transcriptomic changes showed that low dose-rate radiation exposure resulted in changes in gene expression related to cellular metabolic activity, pro-survival and proliferation signaling, alterations in the extracellular matrix and cellular motility, increased expression of factors related to vascular constriction, and increased signaling related to hypoxia and iron deficiency. We confirmed our findings using targeted qPCR and Western blotting in primary HLMVEC. Noticeably absent from the identified DAVID and Metascape pathway analyses were the upregulation of DNA damage responses, strong pro-apoptotic and senescence pathway activation, oxidative stress-related signaling, and marked pro-inflammatory activation pathways, that our laboratory previously identified as major pathway responses to acute exposure to 10 Gy irradiation (1.15 Gy/min) [40]. Together, these data suggest that acute high-dose/high-dose-rate radiation exposure and chronic low dose-rate radiation exposure activate distinct signaling events, resulting in fundamentally different cellular outcomes.

Several previous studies investigated the sensitivity and responses of MSCs to radiation. One study utilized 40–2000 mGy (dose-rate not disclosed), and showed increased apoptosis, senescence, and autophagy, with evidence of DNA damage response at 1–48 h post-irradiation [7]. A second study exposed MSCs to 1–9 Gy radiation at 300 cGy/min, and examined the cells from 12 h to several weeks after irradiation [39]. This study also showed DNA damage response and loss of viability. A third study performed comprehensive analysis of transcriptome changes in MSCs exposed to 0.01 to 1 Gy (0.79 Gy/min), using microarrays to identify pathway regulation over a short time course from 1–48 h post-irradiation [59]. In contrast with our study, which exposed cells to 0.66 cGy/h, the irradiation of the MSCs in all of these studies was performed as acute exposures. Our study did not identify significant DNA damage response, apoptosis or senescence, although it is possible that these processes occurred at a low level earlier in the time course of the exposure, and that a percentage of the remaining cells recovered and proliferated over the time course. Our analysis at the 1- and 2-week time points suggests that the cells continued to proliferate (due to cellular density) and maintained normal (non-senescent) morphology. A previous study using very-low-dose X-ray irradiation of fibroblasts also identified an initial cell cycle pause, followed by resumed proliferative response [60]. The authors concluded that the early pause in cell cycle could have been associated with DNA repair [60]. Future work to investigate very early responses of MSCs to chronic low-dose radiation is needed to determine whether an early response may include a transient pause in the cell cycle.

In our experiments, MSCs responded to chronic low dose-rate radiation by downregulating cholesterol and lipid biosynthesis, as well as downregulating some enzymes in the pathway for oxidative phosphorylation while upregulating enzymes for glycolysis. A shift to aerobic glycolytic metabolism was previously demonstrated in vivo following fractionated high-dose radiation (5 Gy/day × 3 days, 702 cGy/min) [61]. Whether the changes observed in the chronic low-dose exposure is also aerobic or anaerobic glycolysis requires further investigation. With regard to changes in lipid and cholesterol metabolism, a number of reports showed *increased* cholesterol biosynthesis following acute, high-dose radiation exposure, which has been hypothesized to be a potential mechanism for radiation-induced cardiovascular disease and carcinogenesis [62]. In contrast, relatively low-dose radiation (25–50 mGy, 1.0 mGy/min) was shown to reduce atherosclerosis lesions in a murine model predisposed to atherosclerosis, although the mechanism of this is unknown [63].

At both 1 and 2 Gy exposures, we observed changes in the expression of ECM proteins, as well as proteins involved in cellular attachment and motility. The specific alterations in the expression of the ECM proteins and cellular intermediate filaments could have the combined effect of increasing cellular stiffness and increasing motility [64]. In some cases, the changes in proteins that increase cell motility, including some keratins, have also been linked to increased cellular proliferation [65]. Further studies of MSCs following chronic radiation are required to determine whether cellular motility is altered and also to measure specific changes in ECM stiffness.

Although GO analysis of altered gene expression identified a number of altered pathways, analysis of potential pathway hierarchies led us to hypothesize that hypoxia and iron deficiency pathways lie upstream of the other pathways that are regulated. Hypoxia can be associated with increased cellular proliferation and inhibition of apoptosis, in some cases through AQP1 regulation [66]. Additionally, cellular responses to hypoxia, and activation of hypoxia-inducible factors (HIFs), can lead to downstream regulation of cholesterol and lipid metabolism, changes in ECM protein synthesis and cell migration, increased cell survival and proliferation gene expression, and changes in glycolysis [61,67]. In vivo, pulmonary hypoxia leads to decreased cholesterol synthesis, and favors certain types of pulmonary vascular constriction [68]. In agreement with the potent regulation of hypoxia (directly or indirectly), GO analysis identified at least 15 genes known to be activated by hypoxia-inducible factors. Interestingly, we also found *downregulation* of genes associated with oxidative protective mechanisms, including SOD and Nrf2, suggesting that redox stress is not strongly regulated by chronic low dose-rate radiation at the time points that were studied.

Radiation-induced oxygen depletion in aqueous environments was identified over 40 years ago [69,70]. High-dose radiation (15 Gy, 67 cGy/min) induces hypoxia-related gene expression at 24 h post-irradiation [71], and recent advances in FLASH radiotherapy have rekindled interest in this effect [72]. The biological effects of radiation that would lead to cellular iron deficiency or iron-deficiency-like signaling are not known. Although our laboratory has described the effects of radiation on iron in vivo, these effects are initiated by red blood cell and reticulocyte hemolysis [73,74].

We also observed an increase in pathway regulation related to iron deficiency. High levels of iron can inhibit activation of HIFs, whereas low iron can increase HIF activation [75]. Iron deficiency can independently modulate glycolysis, regulate genes involved in cholesterol and lipid metabolism, and increase Egr-1 signaling [76,77]. Studies have shown that hypoxia can affect iron homeostasis and absorption under some conditions [78]. The link between radiation and cellular iron is not known. In cultured breast cancer cells, radiation-induced autophagic cell death was associated with iron accumulation, increased levels of transferrin receptor, and increased ferritin following acute exposure to 1–8 Gy X-ray irradiation (1.0 Gy/min) [79]. One possibility is that autophagy-induced release of cellular iron occurs in chronic low-dose radiation at an early time point, perhaps during an adaptation phase. Such events could result in iron deficiency, but future studies with early time points are needed to determine whether some cellular adaptation occurred early during the irradiation.

Studies of the biological response to low levels of radiation have revealed that there is a large uncertainty in determining actual health risks [80,81]. Several studies have suggested that there may be a non-linear biological response to radiation at low doses, with some data showing complex cellular responses that are not always detrimental [80]. According to the hormesis hypothesis, the dose–response relationship can be non-linear, in which low-dose stress can result in an optimal outcome [82]. Some animal model data from low-dose radiation studies do support a non-linear response [80]. However, the factors that determine in vivo responses to radiation, in animal models or in humans, are not sufficiently understood to predict individual outcomes of low level radiation exposure [35]. Further understanding of pathway regulation by low-dose radiation may allow the identification of markers to predict biological outcomes and also to identify countermeasures for adverse effects of low-dose radiation.

## Figures and Tables

**Figure 1 antioxidants-12-00241-f001:**
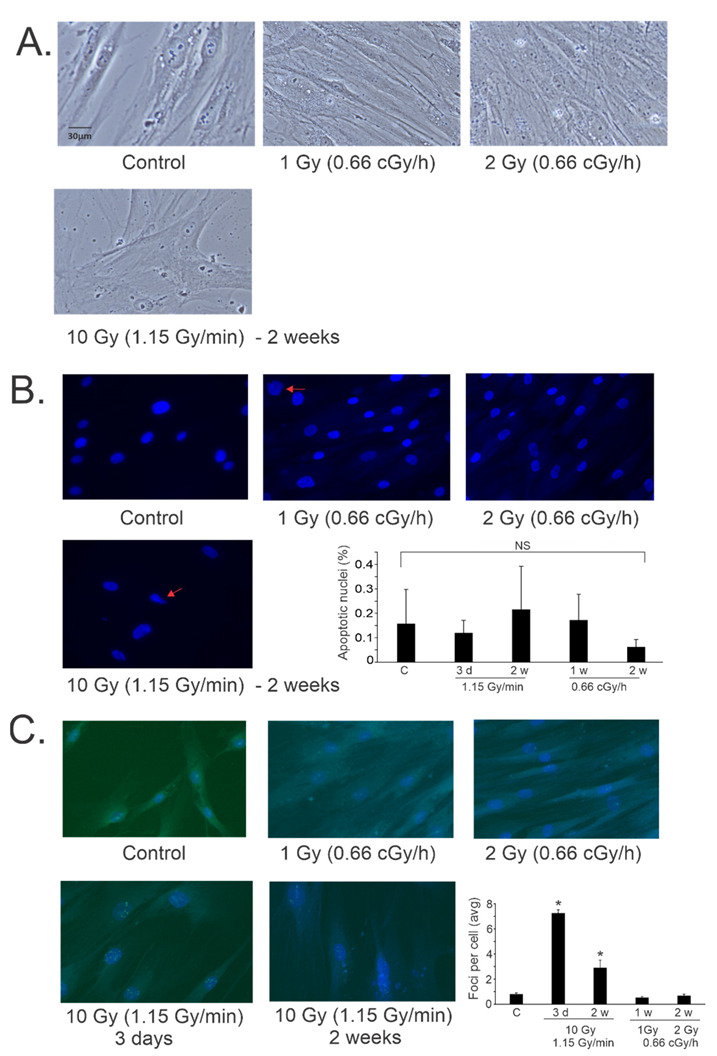
Effects of high and low dose-rate X-ray irradiation on cell morphology, apoptosis, and γ-H2AX foci in MSCs. MSCs were grown to 60% confluence and exposed to ^60^Co irradiation at 0.66 cGy/h for 1 Gy (1 week) or 2 Gy (2 weeks). Alternatively, cells were exposed to 10 Gy X-ray irradiation (1.15 Gy/min) and assayed at 3 days and 1 week post-irradiation. Cells were fixed at the indicated times and stained with DAPI, and immunohistochemistry was performed for γ-H2AX. (**A**). Light microscopy was used to examine cell morphology. Representative images are shown, 20× magnification. (**B**). DAPI was used to examine the nucluear morphology of the fixed cells at the indicated times. Arrows (10 Gy irradiation) indicate nuclear blebbing, a late apoptotic event. Representative images are shown from each condition, 20× magnification. Nuclei were scored from all cells in random fields to determine percentage of apoptotic nuclei at 3 days (3 d), 1 week (1 w) or 2 weeks (2 w) post-irradiation. Graph shows average of percent apoptosis ± SEM; NS = not significant compared with control (C). (**C**). γ-H2AX immunohistochemistry was used to detect foci surrounding double-stranded DNA breaks in the fixed cells at the indicated times. Representative images are shown from each condition, 20× magnification. Foci were scored in all cells from random fields to determine numbers of foci per cell. Graph shows average of nuclear foci ± SEM; * indicates *p* < 0.05 compared with sham-irradiated control cells.

**Figure 2 antioxidants-12-00241-f002:**
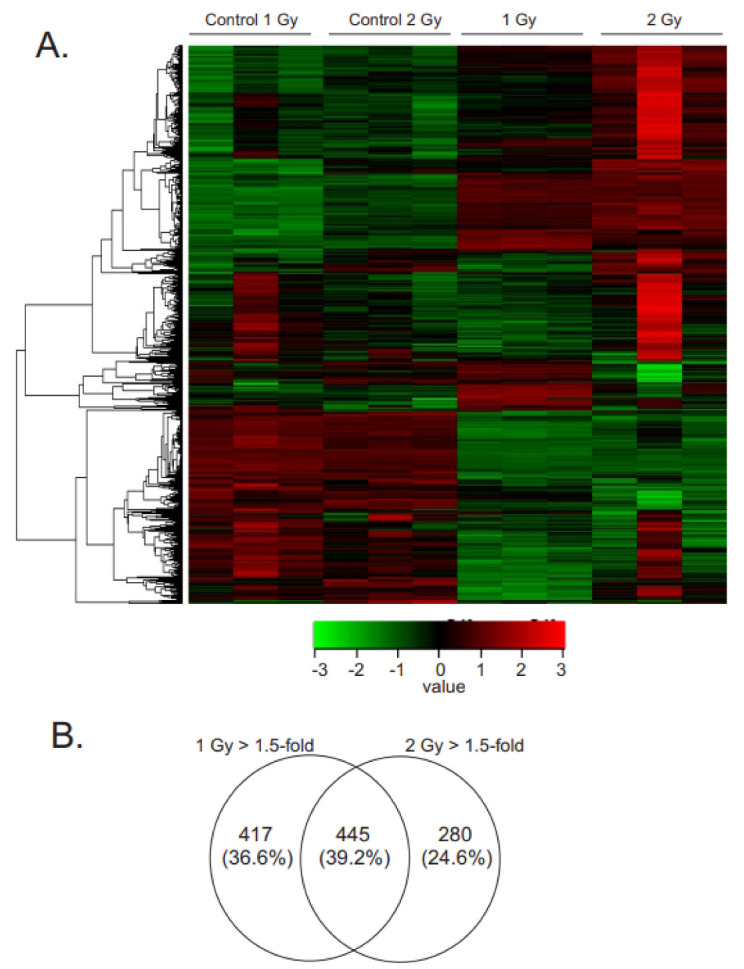
Gene expression changes in irradiated MSCs. MSCs were grown to 60% confluence and exposed to ^60^Co irradiation at 0.66 cGy/h for 1 Gy (1 week) or 2 Gy (2 weeks). Control cells were cultured under identical conditions for each time point. RNAseq was performed using N = 3 samples for each condition. (**A**). Heat map indicates gene expression patterns following radiation exposure. (**B**). Venn diagram illustrating the number of genes with altered expression at each time point, q < 0.05, absolute fold change >1.5.

**Figure 3 antioxidants-12-00241-f003:**
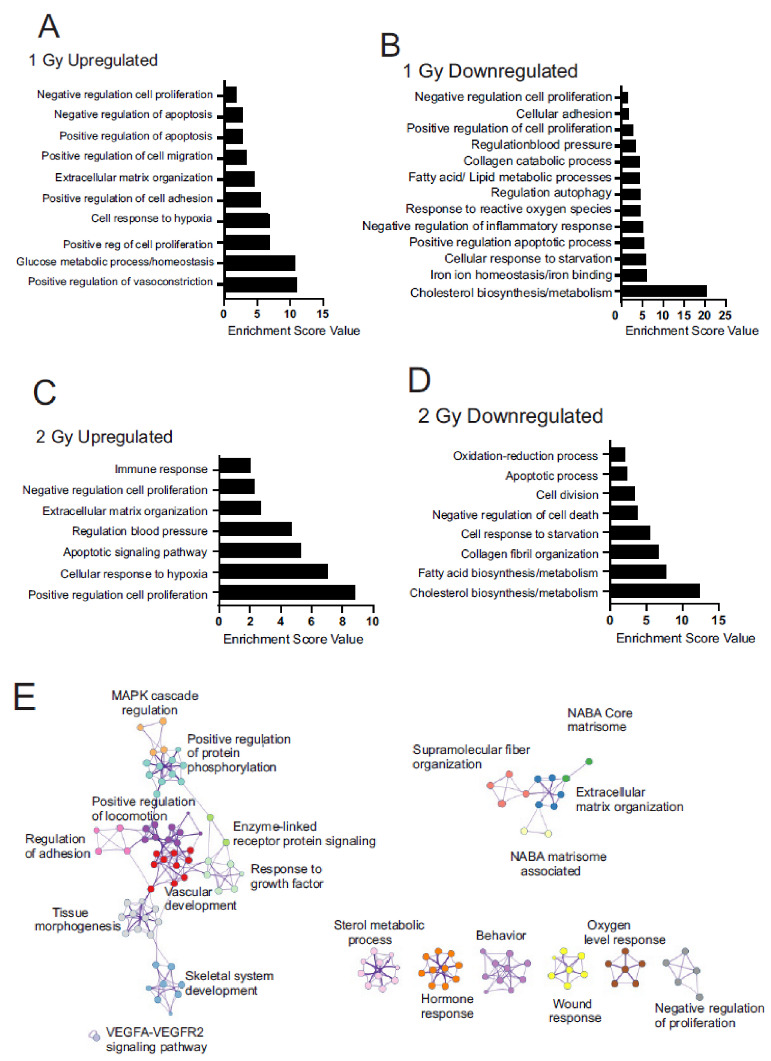
GO term cluster, KEGG pathway enrichment, and Metascape analyses of differentially expressed genes in MSCs following chronic low dose-rate irradiation. MSCs were grown to 60% confluence and exposed to ^60^Co irradiation at 0.66 cGy/h for 1 Gy (1 week) or 2 Gy (2 weeks). Control cells were cultured under identical conditions for 1 or 2 weeks. Irradiated and control cells were lysed at the same time, and RNA was prepared for RNAseq. Pathway regulation was compared for all conditions. A,B. 1 Gy irradiation, upregulated pathways (**A**) and downregulated pathways (**B**). C,D. 2 Gy irradiation, upregulated pathways (**C**) and downregulated pathways (**D**). (**E**). Clustered GO terms using Metascape using genes with q < 0.05 to visualize pathway relationships.

**Figure 4 antioxidants-12-00241-f004:**
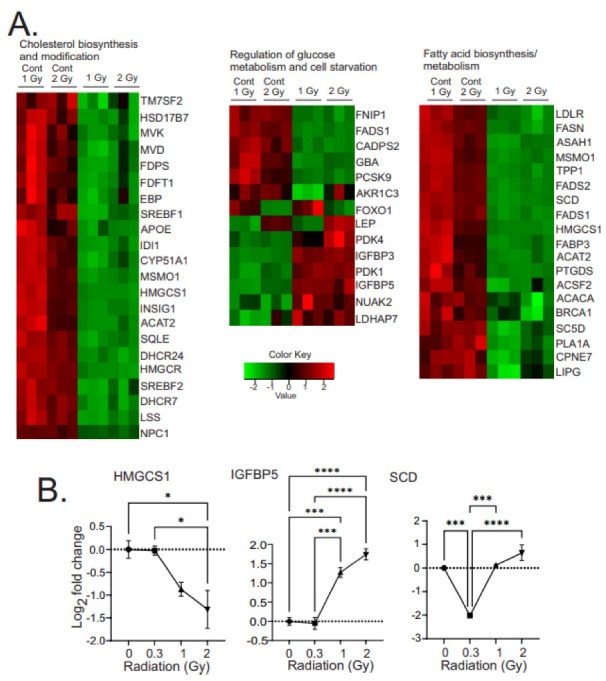
Heatmaps of gene expression changes in pathways for cholesterol biosynthesis and modification, glucose metabolism and cell starvation, and fatty acid biosynthesis and metabolism. (**A**). RNAseq was used to identify gene expression changes in primary human MSCs following 1 or 2 Gy (0.66 cGy/h) ^60^Co irradiation. Control cells were cultured under identical conditions for 1 or 2 weeks. GO analysis was performed using DAVID and heatmaps were generated using genes with q < 0.05. Rows are centered and unit variance is applied to rows. (**B**). qPCR gene regulation using log base 2 scale fold change of genes supporting heatmaps of pathways. Data show averages ± SEM N = 3 biological replicates with two technical repeats; * indicates *p* < 0.05, *** indicates *p* < 0.001, **** indicates *p* < 0.0001, respectively, compared with sham irradiated control cells.

**Figure 5 antioxidants-12-00241-f005:**
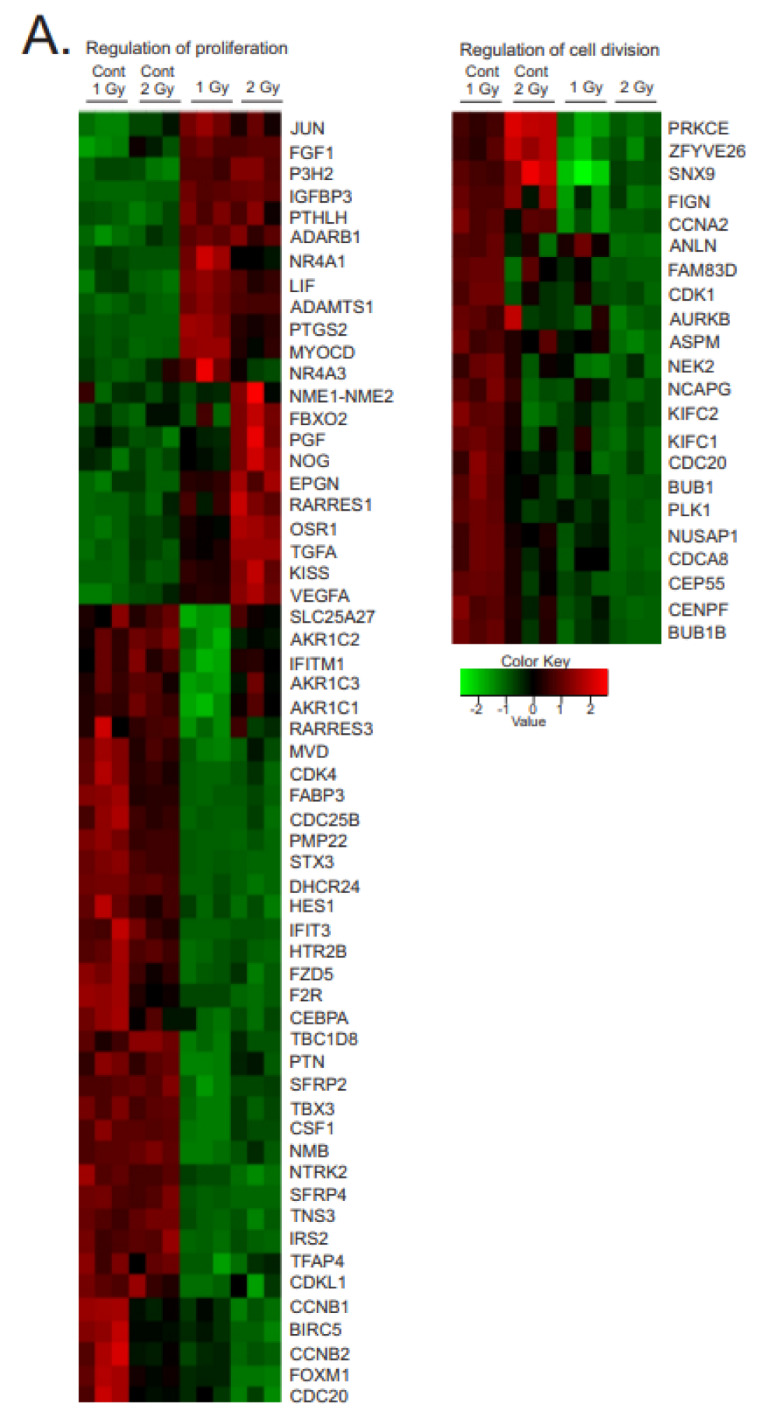
**Heatmaps of gene expression changes for pathways in regulation of proliferation and cell division.** (**A**). RNAseq was used to identify gene expression changes in primary human MSCs following 1 or 2 Gy (0.66 cGy/h) ^60^Co irradiation. Control cells were cultured under identical conditions for 1 or 2 weeks. GO analysis was performed using DAVID and heatmaps were generated using genes with q < 0.05. Rows are centered and unit variance is applied to rows. (**B**). qPCR gene regulation using log base 2 scale fold change of genes supporting heatmaps of pathways. Data show averages ± SEM N = 3 biological replicates with two technical repeats; * indicates *p* < 0.05, ** indicates *p* < 0.01, *** indicates *p* < 0.001, respectively, compared with sham-irradiated control cells. (**C**). Western blot data showing regulation of AKT and MAPK (phosphorylated and total). Western blots were performed on N = 3 biological repeats. Bar graphs show average band densities normalized to β-actin. Graphs show means ± SEM; * indicates *p* < 0.05 and ** indicates *p* < 0.01, respectively, compared with sham-irradiated control cells.

**Figure 6 antioxidants-12-00241-f006:**
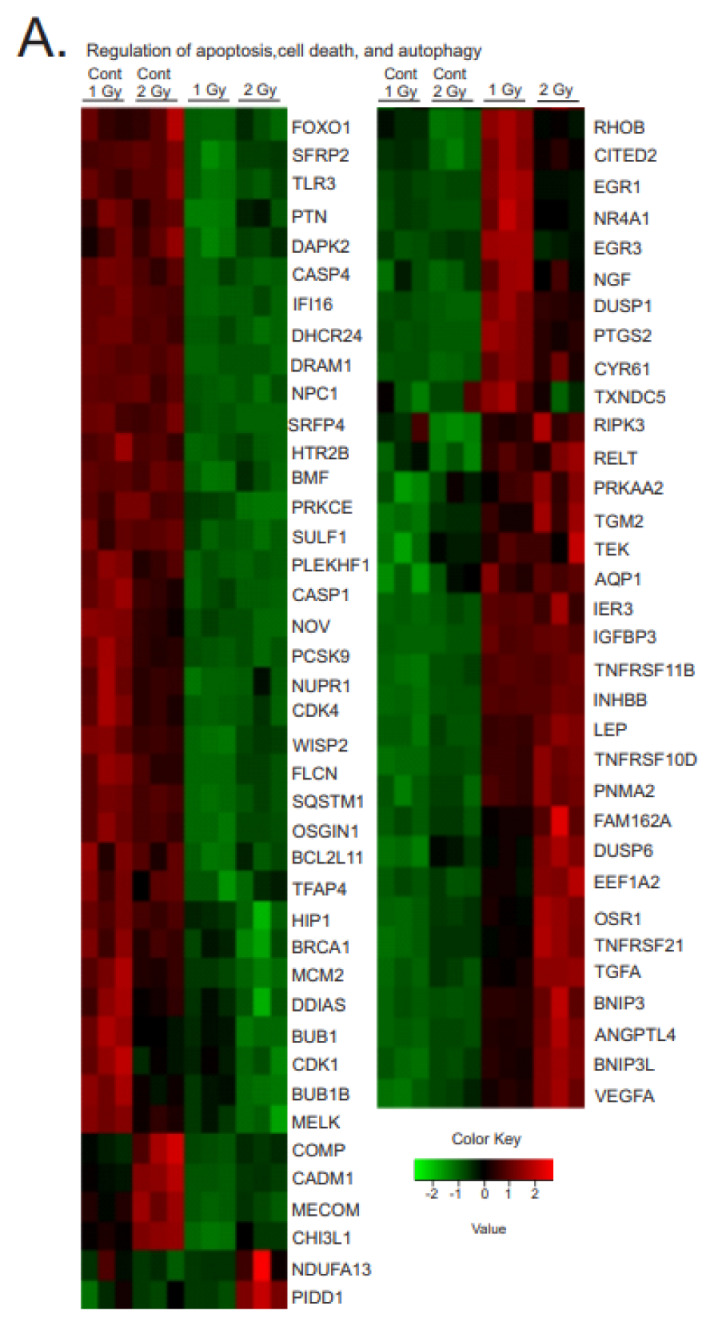
**Heatmaps of gene expression changes for pathways in regulation of apoptosis, cell death, and autophagy.** (**A**). RNAseq was used to identify gene expression changes in primary human MSCs following 1 or 2 Gy (0.66 cGy/h) ^60^Co irradiation. Control cells were cultured under identical conditions for 1 or 2 weeks. GO analysis was performed using DAVID and heatmaps were generated using genes with q < 0.05. Rows are centered and unit variance is applied to rows. (**B**). qPCR gene regulation using log base 2 scale fold change of genes supporting heatmaps of pathways. Data show averages ± SEM from N = 3 biological replicates with two technical repeats. * indicates *p* < 0.05, ** indicates *p* < 0.01, *** indicates *p* < 0.001, **** indicates *p* < 0.0001, respectively, compared with sham-irradiated control cells. (**C**). Western blot data showing regulation of Egr1 and DUSP1. Western blots were performed on N = 3 biological repeats. Bar graphs show average band densities normalized to β-actin.

**Figure 7 antioxidants-12-00241-f007:**
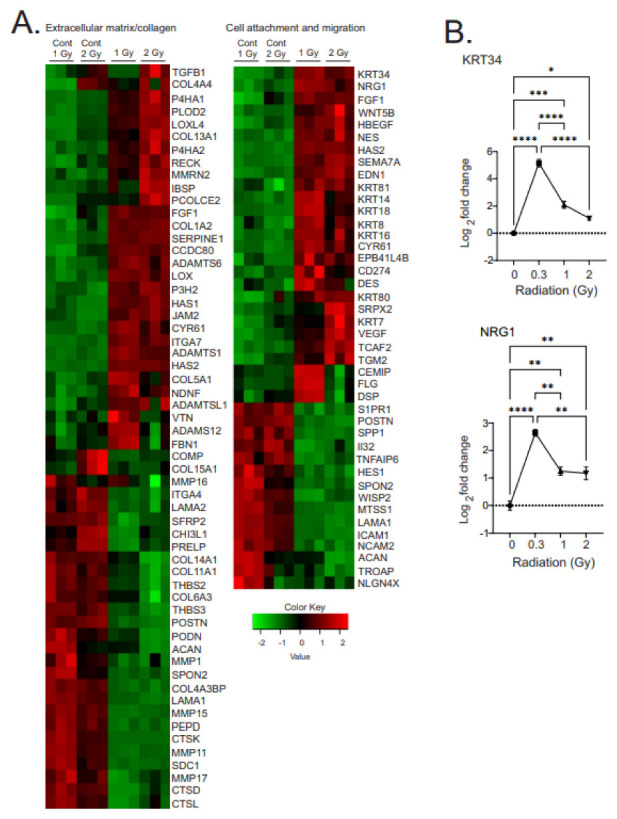
Heatmaps of gene expression changes for pathways involved in the regulation of extracellular matrix and collagen synthesis and cell attachment and migration. (**A**). RNAseq was used to identify gene expression changes in primary human MSCs following 1 or 2 Gy (0.66 cGy/h) ^60^Co irradiation. Control cells were cultured under identical conditions for 1 or 2 weeks. GO analysis was performed using DAVID and heatmaps were generated using genes with q < 0.05. Rows are centered and unit variance is applied to rows. (**B**). qPCR gene regulation using log base 2 scale fold change of genes supporting heatmaps of pathways. Data show averages ± SEM from N = 3 biological replicates with two technical repeats. * indicates *p* < 0.05, ** indicates *p* < 0.01, *** indicates *p* < 0.001, **** indicates *p* < 0.0001, respectively, compared with sham-irradiated control cells.

**Figure 8 antioxidants-12-00241-f008:**
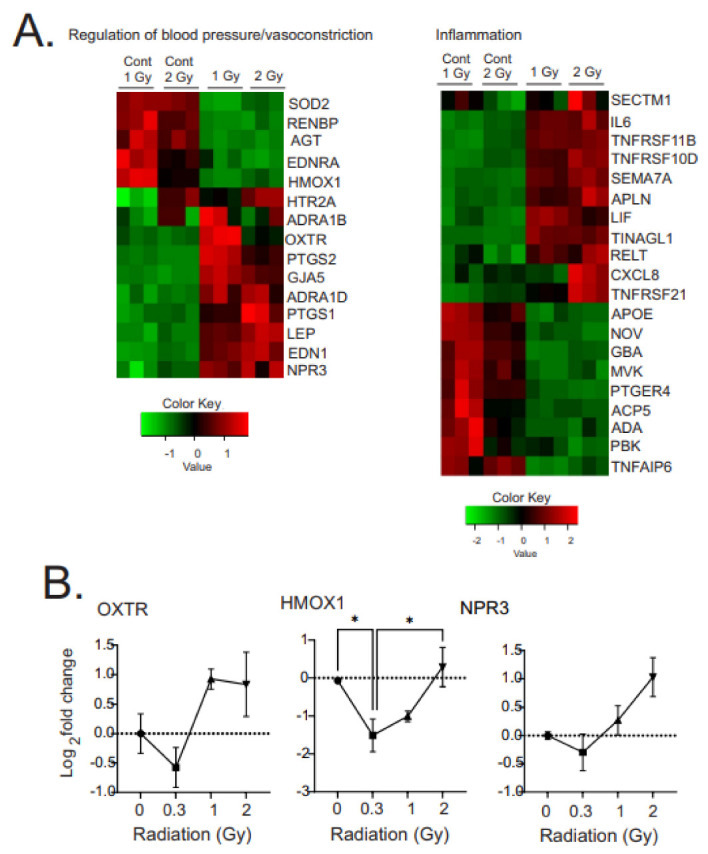
Heatmaps of gene expression changes for pathways involved in the regulation of blood pressure or vasoconstriction and inflammation. (**A**). RNAseq was used to identify gene expression changes in primary human MSCs following 1 or 2 Gy (0.66 cGy/h) ^60^Co irradiation. Control cells were cultured under identical conditions for 1 or 2 weeks. GO analysis was performed using DAVID and heatmaps were generated using genes with q < 0.05. Rows are centered and unit variance is applied to rows. (**B**). qPCR gene regulation using log base 2 scale fold change of genes supporting heatmaps of pathways. Data show averages ± SEM from N = 3 biological replicates with two technical repeats. * indicates *p* < 0.05 compared with sham-irradiated control cells.

**Figure 9 antioxidants-12-00241-f009:**
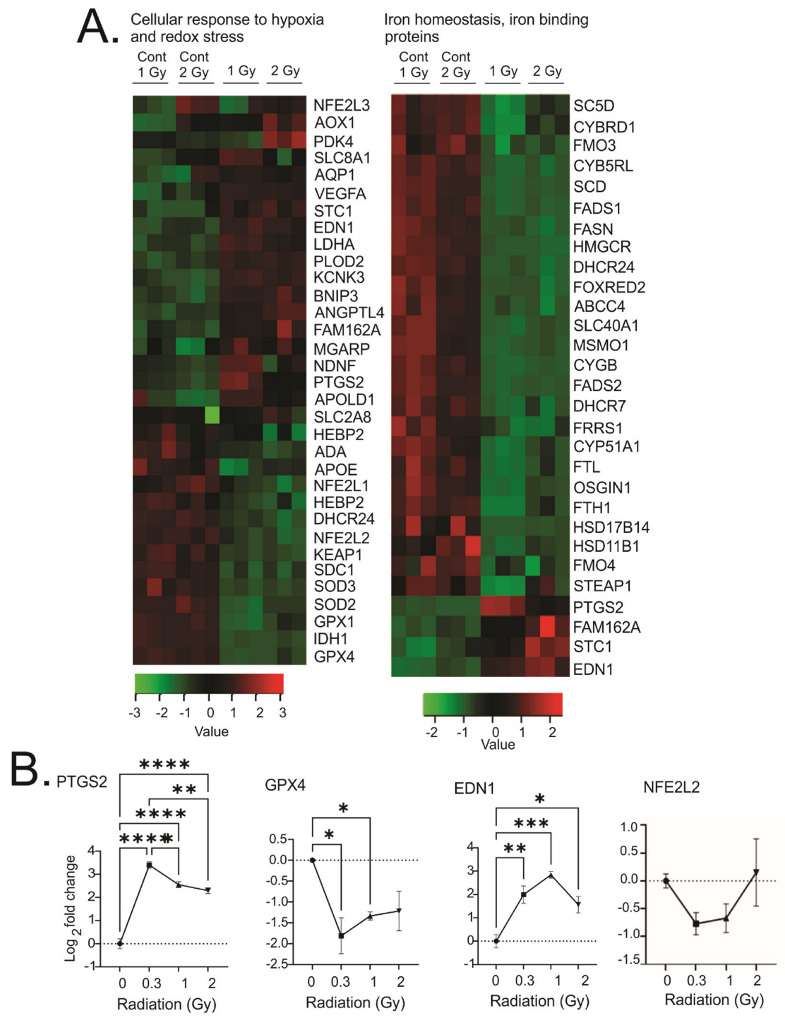
Heatmaps of gene expression changes for pathways involved in cellular response to hypoxia or redox stress and iron homeostasis and iron-binding proteins. (**A**). RNAseq was used to identify gene expression changes in primary human MSCs following 1 or 2 Gy (0.66 cGy/h) ^60^Co irradiation. Control cells were cultured under identical conditions for 1 or 2 weeks. GO analysis was performed using DAVID and heatmaps were generated using genes with q < 0.05. Rows are centered and unit variance is applied to rows. (**B**). qPCR gene regulation using log base 2 scale fold change of genes supporting heatmaps of pathways. Data show averages ± SEM from N = 3 biological replicates with two technical repeats. * indicates *p* < 0.05, ** indicates *p* < 0.01, *** indicates *p* < 0.001, **** indicates *p* < 0.0001, respectively, compared with sham-irradiated control cells.

**Figure 10 antioxidants-12-00241-f010:**
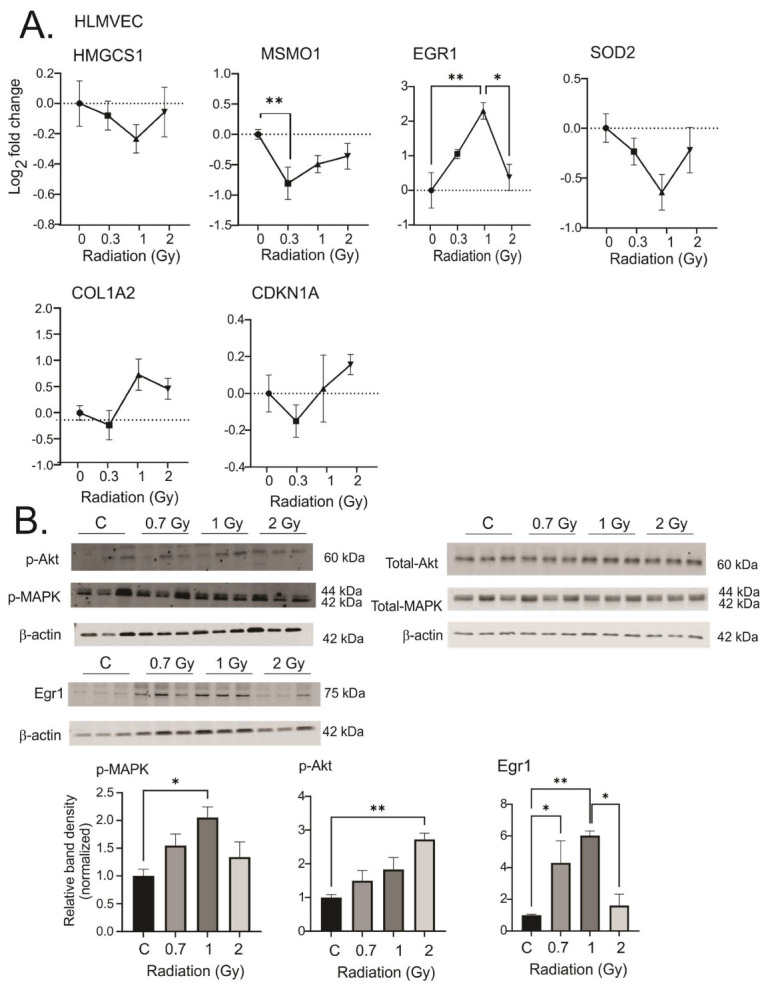
Response of human lung microvascular endothelial cells (HLMVECs) to chronic low dose-rate irradiation. HLMVECs were grown to 50% confluence and exposed to ^60^Co irradiation at 0.64 cGy/h for 1 Gy (1 week) or 2 Gy (2 weeks). Control cells were cultured under identical conditions. Irradiated and control cells were lysed at the same time, and either frozen for protein analysis or placed in RNAlater. (**A**). qPCR gene regulations are represented on a log base 2 fold change scale, using N = 3 biological replicates with two technical repeats. Graphs show means ± SEM; * indicates *p* < 0.05 and ** indicates *p* < 0.01, respectively, compared with sham-irradiated control cells. (**B**). Western blots of phosphorylated and total MAPK and Akt, and Egr1. Western blots were performed on N = 3 biological repeats. Bar graphs show average band densities normalized to β-actin. Graphs show means ± SEM; * indicates *p* < 0.05 and ** indicates *p* < 0.01, respectively, compared with sham-irradiated control cells.

**Table 1 antioxidants-12-00241-t001:** Primers for qPCR.

HGNC Gene Symbol	Forward Primer	Reverse Primer
GAPDH	5′-AGCCACATCGCTCAGACAC-3′	5′-GCCCAATACGACCAAATCC-3′
HMGCS1	5′-TTGTGCCCGAAGGAGGAAAC-3′	5′-CTGGCCCAAGCCAATGGTAT-3′
PTGS2	5′-CTGATGATTGCCCGACTCCC-3′	5′-CGCAGTTTACGCTGTCTAGC-3′
GTPBP4	5′-GAAAATTACGGTGGTGCCGTC-3′	5′-GCCCCAGAGCCAACTTGTA-3′
EGR1	5′-CCCCGACTACCTGTTTCCAC-3′	5′-TGGGTTTGATGAGCTGGGAC-3′
DUSP1	5′- CAGAGCCCCATTACGACCTC-3′	5′-TTGGTCCCGAATGTGCTGAG-3′
KRT34	5′-TCAGAAGCAAGTACCAGACGGA-3′	5′-CTGACTCCTGGTCTCGTTCAG-3′
NRG1	5′-CTGGTGATCGCTGCCAAAAC-3′	5′-GTAGGCCACCACACACATGA-3′
CLCA2	5′-ACTGTGGGCAACGACACTATG-3′	5′-TTCAGGGTGTAAGTCCAGTGC-3′
END1	5′-CTGCCTTTTCTCCCCGTTAAA-3′	5′-GGACTGGGAGTGGGTTTCTC-3′
OXTR	5′-TCCTGTACCCATCCAGCGA-3′	5′-TCCGCAGGCGAACCTAAAG-3′
NPR3	5′-CTGAGTACTCGCACCTCACG-3′	5′-TCACTGCTCGCACACATGAT-3′
LIPG	5′-AGTTGTGGTTGACTGGCTCC-3′	5′-TGTGATTGCTGTGATTCGGC-3′
ACSF2	5′-TTCAGTTCCCAGTAGCTTCACT-3′	5′-CTCCTTGAGTTGGGCAAAGGT-3′
LOXL4	5′-TCTGCGGATCACATGGACTG-3′	5′-AAAGTTGGCACATGCGTAGC-3′
HAS2	5′-TCCCGGTGAGACAGATGAGT-3′	5′-GGCTGGGTCAAGCATAGTGT-3′
COL1A2	5′-TGTGGATACGCGGACTTTGT-3′	5′-CAGCAAAGTTCCCACCGAGA-3′
CYBRD1	5′-AGGGCATCGCTTCTTTCAGGTTT-3′	5′-ACGAAAACACCTTCTGGCGG-3′
ADA	5′-GGAACCAGGCTGAACTGGTC-3′	5′-GCCGCTCTGTCTTGAGTATGT-3′
SOD2	5′-CTGTTGGTGTCCAAGGCTCA-3′	5′-GTAGTAAGCGTGCTCCCACA-3′
PTGER4	5′-CGCTCGTGGTGCGAGTATT-3′	5′-GGGAGATGAAGGAGCGAGAGT-3′
GPX4	5′-GCCTTTGCCGCCTACTGA-3′	5′-CTTGGCGGAAAACTCGTGC-3′
HMOX1	5′-TGCGTTCCTGCTCAACATCC-3′	5′-AGTGTAAGGACCCATCGGAGA-3′
TGFB1	5′-TGGACATCAACGGGTTCACT-3′	5′-GAAGTTGGCATGGTAGCCCT-3′
IL6	5′-TCCTTCTCCACAAACATGTAACAA-3′	5′-TCACCAGGCAAGTCTCCTCA -3′
APOE	5′-GGGGCCTCTAGAAAGAGCTGG-3′	5′-TAATCCCAAAAGCGACCCAGT-3′
MSMO1	5′-GGTTCCGAGGTTGGAACACCT-3′	5′-TTCAAATCTCTGCAGACAGCCT-3′
KISS1	5′-CCACTTTGGGGAGCCATTAGA-3′	5′-CAGTTGTAGTTCGGCAGGTC-3′
NFE2L2	5′-TTCGGCTACGTTTCAGTCAC-3′	5′-TGTCCTGTTGCATACCGTCT-3′

Gene ID numbers: GAPDH #2597; HMGCS1 #3157; PTGS2 #5743; GTPBP4 #23560; EGR1 #1958; DUSP1 #1843; KRT34 #3885; NRG1 #3084; CLCA2 #9635; END1 #55823; OXTR #5021; NPR3 #4883; LIPG #9388; ACSF2 #80221; LOXL4 #84171; HAS2 #3037; COL1A2 #1278; CYBRD1 #79901; ADA #100; SOD2 #6648; PTGER4 #5734; GPX4 #2879; HMOX2 #3162; TGFB1 #7040; IL6 #3569; APOE #348; MSMO1 #6307; KISS1 #3814; NFE2L2 #4780.

## Data Availability

All transcriptomics data will be made available upon request.

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
