# Peer review of "Transcriptomic Profiling and Pathway Analysis of Mesenchymal Stem Cells Following Low Dose-Rate Radiation Exposure"

_antioxidants, 2023, doi:10.3390/antiox12020241_

Round 1
Reviewer 1 Report
Radiation cancer risk has been intensively investigated, even at low dose rates of environmental or industrial radiation. However, sufficient data directly linked to low dose rate radiation risk assessment have yet to be obtained. The research findings are important for deepening the general public's understanding of radiation. This paper investigated changes in gene expression following chronic low-dose rate radiation in mesenchymal stem cells. Overall, the manuscript is well written. However, there is concern about data presentation and the lack of proper explanation. Authors should address this reviewer’s comments which are described below.
Major issues
1. According to UNSCEAR Report, low dose is defined below 100 mSv. This studied did not use low-dose radiation but used low-dose rate. Thus, manuscript title (low dos radiation exposure) is incorrect.
2. Data quality of images in Figure 1A, 1B, 1C is poor. In order to make them clearer, change the resolution of an image or magnify it.
3. Describe how to count the number of cells with gamma-H2AX foci.
Minor issue
1. Line 394, Amp should be AMP.
Author Response
Major issues
- According to UNSCEAR Report, low dose is defined below 100 mSv. This studied did not use low-dose radiation but used low-dose rate. Thus, manuscript title (low dose radiation exposure) is incorrect. We have corrected the title of the manuscript, and other instances in the manuscript to clarify that this is low dose-rate
- Data quality of images in Figure 1A, 1B, 1C is poor. In order to make them clearer, change the resolution of an image or magnify it. Improved images of the cell are now provided in Fig 1.
- Describe how to count the number of cells with gamma-H2AX foci. Immunoreactive γ-H2AX foci were counted in 100 DAPI-positive cells per slide. The images were taken using the same exposure conditions for all of the slides. We have tried to update this in the Methods section.
Minor issue
- Line 394, Amp should be AMP. This has been corrected on page 13.
Reviewer 2 Report
The present manuscript has comprehensively described the article about the
transcriptomic genetic profiles and related pathways of the effect of low-dose radiation on human mesenchymal stem cells. The further Gene ontology
analysis demonstrated the alterations in multiple pathways related to cellular metabolism, extracellular matrix modification, cell adhesion/migration, regulation of vasoconstriction and inflammation, and cell response to hypoxia and iron hemostasis.
However, there are a few questions that should be clarified:
1. The presented gene difference after exposure to low-dose radiation in human mesenchymal stem cells is the result of in vitro experiments without interaction of tumor microenvironment. The results should be validated carefully in further in vivo studies.
2. The quality and resolution of Fig. 1 should be revised.
3. In the abstract, the authors emphasized the finding of a reduction of Nrf2. However, there are no any verified qPCR results in the presented data.
Author Response
- The presented gene difference after exposure to low-dose radiation in human mesenchymal stem cells is the result of in vitro experiments without interaction of tumor microenvironment. The results should be validated carefully in further in vivo studies. Our goal in this study was to determine the effects of low dose-rate radiation on normal tissues. We agree that the response of tumors and cells in a tumor microenvironment might differ significantly.
- The quality and resolution of Fig. 1 should be revised. We will provide improved images for Fig. 1 in the revised manuscript.
- In the abstract, the authors emphasized the finding of a reduction of Nrf2. However, there are no any verified qPCR results in the presented data. We have performed qPCR to validate the downregulation of Nrf2. We find that it is strongly downregulated at 0.3 and 1 Gy in the qPCR and then returns to near baseline level at 2 Gy. These data are now included in Fig 9B. We hope to follow this up in the future.